# Hunting Dynamics and Identification of Potentially Pathogenic Bacteria in European Fallow Deer (*Dama dama*) across Three Hunting Reserves in Western Romania

**DOI:** 10.3390/microorganisms12061236

**Published:** 2024-06-19

**Authors:** Iulia-Maria Bucur, Alex Cristian Moza, Mirel Pop, Ileana Nichita, Cristina Mirabela Gaspar, Răzvan Cojocaru, Radu-Valentin Gros, Marius Valentin Boldea, Andreea Tirziu, Emil Tirziu

**Affiliations:** 1Faculty of Veterinary Medicine, University of Life Sciences “King Mihai I” from Timisoara, Calea Aradului 119, 300645 Timisoara, Romania; bucur_iulia@ymail.com (I.-M.B.); mirelvet@yahoo.com (M.P.); nichita_ileana@yahoo.com (I.N.); gasparcristina99@yahoo.com (C.M.G.); razvan.cojocaru@usvt.ro (R.C.); grosrv@yahoo.com (R.-V.G.); marius_boldea@usab-tm.ro (M.V.B.); emiltarziu@yahoo.com (E.T.); 2Faculty of Medicine, “Victor Babes” University of Medicine and Pharmacy, Piata Eftimie Murgu 2, 300041 Timisoara, Romania; andreea.tirziu@yahoo.com

**Keywords:** *Dama dama*, fallow deer, harvesting, dynamics, zoonotic diseases, bacterial pathogens

## Abstract

The study focused on the hunting practices and potentially pathogenic bacterial species among European fallow deer (*Dama dama)*. Within a five-year period, three hunting grounds from Western Romania were examined. During this period, a total of 1881 deer were hunted, and 240 samples were collected by rectal and nasal swabbing from 120 carcasses. Bacterial strains were identified utilizing bacteriological assays and the Vitek^®^ 2 Compact system. Notably, the Socodor hunting ground exhibited a significant difference in harvesting quotas between the bucks (Group M) and does/yearlings (Group F), favoring the latter. In the Chișineu Criș–Sălișteanca hunting ground, a likely correlation in harvesting quotas between the two groups was observed. The identified potentially pathogenic bacteria were *Escherichia coli*, *Salmonella* spp., *Staphylococcus aureus*, *Listeria monocytogenes* and *Enterococcus faecium*. These results highlight the importance of effectively managing the deer population and recognize the potential for *Dama dama* to spread zoonotic pathogens, emphasizing the necessity of adopting a One Health approach and maintaining ongoing surveillance of this game species’ population dynamics.

## 1. Introduction

The European fallow deer (*Dama dama*) roams freely across Europe’s various habitats, representing one of the most widespread cervid species worldwide [1]. The expansion of the fallow deer’s range can be attributed to deliberate translocation and introduction to new areas, motivated by pursuits such as trophy hunting, the demand for game meat, or aesthetic reason [2].

According to recent official data of Romanian legislation, the fallow deer population stands at 7508 individuals, from which a harvest rate of approximately 21.2% is approved for the ongoing year. Despite this management approach, the species is classified as “Least Concern” on The International Union for Conservation of Nature (IUCN) Red List of Threatened Species and thus is not an endangered species [3]. On the contrary, *Dama dama* is protected by the Law on Hunting and Protection of Hunting Funds, and the Law for Romania’s Accession to the Convention on the Conservation of Wildlife and Natural Habitats in Europe [4,5]. However, the management and harvesting of wild ungulates have different approaches across the European countries [6,7].

The fallow deer serves as a considerable game species throughout Europe, valued for both its ecological contributions and economic importance in hunting activities [8,9]. Based on recent statistics, the population of *Dama dama* in Europe is estimated at approximately 951,521 specimens, showing an almost seven-fold increase from 1984 to 2020. Simultaneously, the deer harvest increased six-fold over the same period, reaching a maximum point of 202,896 animals in 2020 [1]. 

Wildlife is involved in upholding ecological balance and biodiversity, and the link between humans, animals, and the environment is also being recognized by One Health [10]. In addition, public health is influenced by the health of animals, considering the multitude of unidentified microorganisms that wildlife may carry [11].

Nevertheless, concerns emerge regarding the potential role of the fallow deer as a source for pathogenic bacteria, presenting a risk to humans and as well to domestic and wild animals. The rise of these pathogenic microbes is a consequence of anthropogenic activities, climate change, biodiversity loss, habitat degradation, and the rising rate of wildlife–human interaction [11,12,13]. These concerns are observed, especially with an increasing interest in the research of wildlife populations and the bacterial communities harbored by them, with a focus on the fecal and intestinal microbiota of cervids [14]. 

While numerous studies have investigated the bacterial populations in various wildlife species [15,16], limited research has focused on the nasal and intestinal bacterial flora of European fallow deer that could eventually give rise to contaminated game meat [17,18], particularly in the regions where hunting activities are prevalent, such as Romania.

Notable studies have pointed out the prevalence of pathogenic bacteria, including *Escherichia coli*, *Salmonella* spp., *Staphylococcus aureus*, *Listeria monocytogenes*, *Campylobacter* spp., *Yersinia enterocolitica*, *Yersinia pseudotuberculosis* or *Mycobacterium bovis*, within wild deer populations in Europe. This highlights the need for more comprehensive investigations to prevent the public health risks linked to zoonotic diseases or game consumption [15,18,19,20,21,22].

Overall, the lack of information shows the need for consistent and reliable data collection, especially through monitoring programs, in order to effectively manage the rising fallow deer populations as a renewable natural resource, but also a highly competent reservoir of potentially zoonotic bacteria.

In this regard, this study aimed to address this knowledge gap by analyzing the hunting dynamics of wild *Dama dama* across three hunting grounds in Western Romania, over a five-year span. Furthermore, our findings will provide valuable insights into microbial populations and the incidence of potentially pathogenic bacteria among the fallow deer. Ultimately, this research has implications for public health, wildlife management, and food safety policies. 

## 2. Materials and Methods

### 2.1. Harvesting Quota Dynamics and Sample Collection

Between 2017 and 2021, a considerable number of European fallow deer (*n* = 1881) were hunted during organized hunting activities, within three of the most important hunting grounds in the Arad and Timiș counties (Western Romania). The hunting locations are called Socodor (A), harboring the largest fallow deer population in Romania [23], Chișineu Criș–Sălișteanca (B), and Nadăș (C) (Figure 1). The harvesting quota was documented annually for each of the three hunting reserves. Thus, samples were collected and analyzed from a total of 120 animals.

Each animal was examined morphologically prior to sample collection in order to define the species and sex. Afterwards, 240 samples taken by both rectal and nasal swabbing were obtained from the carcasses within a one-hour window following the death of the animals, resulting in a total of 240 samples. It should be noted that all the samples were collected using sterile dry cotton swabs placed in polypropylene tubes, without a transport medium. Veterinary practitioners performed the sampling immediately after the carcasses had been harvested. The swabs were stored at refrigeration temperatures (2 to 4 °C) in a cooling box and transported within 1 to 3 h to the Faculty of Veterinary Medicine in Timișoara, where they were subsequently screened for potentially pathogenic bacteria upon arrival.

### 2.2. Bacterial Preliminary Isolation and Classification

The collected samples (*n* = 240) underwent classic bacteriological techniques in order to isolate different species of bacteria. Initially, the swabs were aseptically transferred in tubes containing Nutrient Broth and incubated for 24 h at 37 °C. Then, in order to promote the growth of different species, bacteriological cultures were carried out using selective and differential culture media. In this regard, the Eosin methylene blue agar (also known as EMB, or Levine agar) and Propylene glycol deoxycholate neutral red agar (also known as Rambach) were used for the isolation of Gram-negative bacteria, while Listeria selective agar base (Oxford formulation) and Mannitol salt agar (also known as MSA, or Chapman agar) were used for the isolation of Gram-positive bacteria. Additionally, the most significant colonies were microscopically examined via smear preparations and subsequently Gram staining [24,25,26].

### 2.3. Bacterial Biochemical Identification

Individual colonies that had grown on the surface of the selective media were carefully transferred onto blood agar plates. This process was performed in order to cultivate the pure 24 h cultures needed for bacterial identification with the advanced Vitek^®^ 2 Compact system (bioMérieux, Craponne, France). For this, identification cards were used for both Gram-negative (Vitek^®^ 2 GN) and Gram-positive bacteria (Vitek^®^ 2 GP).

## 3. Results

### 3.1. Harvesting Quota Dynamics and Sample Collection

In a five-year period, between 2017 and 2021, 1881 specimens of fallow deer were hunted. The categories of deer were bucks, does, and yearlings under the age of 2 years. Harvesting quotas were registered for each year in all three hunting grounds, Socodor (A), Chișineu Criș–Sălișteanca (B) and Nadăș (C), to determine the local hunting dynamics of this wild species. Thus, the deer were classified in two groups as follows: Group M for the bucks and Group F for the does and yearlings (Table 1). It should be mentioned that, regarding hunting ground C, the harvesting quotas were insufficient to run a statistically significant analysis.

Based on these data, the harvesting quota in the hunting ground of Socodor (A), with the largest population of European fallow deer in this study, had a slight decrease regarding group F from 2020 onwards. On the other side, group M presented a constant growth in harvesting quota every year. Maximum values were reached in 2019 by group F (*n* = 245), and by group M in 2021 (*n* = 130) (Figure 2).

Regarding hunting ground Chișineu Criș–Sălișteanca (B), the harvest quota suffered a slight decrease in the beginning, reaching a minimum point (*n* = 20) in 2019 and 2020 for group M and group F (*n* = 32), respectively. Nonetheless, in the last year, a new maximum harvest quota was recorded in both groups, M (*n* = 25) and F (*n* = 38), as can be observed in Figure 3.

Lastly, in hunting ground C, the harvesting data were insufficient to run a statistically significant analysis. However, despite the lack of data, a constant increase in the harvesting quota was observed during the period of study, starting from a minimum of 20 deer in 2017, and reaching a maximum of 40 deer in 2021 (Table 1).

By analyzing the data, it can be seen that for location A, a negative correlation was found, with no statistically significance (r(8) = −0.33; *p* > 0.05) between the dynamics of groups M and F harvesting quotas, as indicated by a low coefficient of determination (R^2^ = 0.1125) (Figure 3). On the other hand, in the case of hunting ground B, a strong positive correlation between the hunting dynamics of the two groups was found (r(8) = 0.95; *p* < 0.05; R^2^ = 0.9124) (Figure 4).

A limitation to this study would be not including numerical analyses of the local living population of fallow deer through the period of study, due to the lack of data for the desired period [27].

Throughout the study, 120 European fallow deer carcasses were selected at random and subjected to sampling through rectal and nasal swabs, reaching a total of 240 samples. The carcasses were subjected to an autopsy, which displayed evidence of traumatic injuries consistent with gunshot wounds, characterized by entrance and exit wounds, tissue disruption, and hemorrhage indicative of ballistic trauma. None of the sampled carcasses presented lesions that could suggest that the specimens were suffering from infectious diseases. Thereafter, the samples, represented by rectal and nasal swabs, underwent screening for potentially pathogenic bacteria.

### 3.2. Bacterial Preliminary Isolation and Classification

The bacteriological examinations revealed that out of 240 samples, 221 (92.08%) were positive for bacteria, developing colonies on the culture media. For the used culture media, the frequency of isolation was the highest on the Chapman agar (96/240, 40%) and the Levine agar (93/240, 38.75%), followed by the Rambach (24/240, 10%) and Oxford (8/240, 3.33) Agars (Table 2). Following the Gram staining method, it was noticed that amongst the 221 examined isolates, 117 (52.94%) had Gram-negative morphology, while 104 (47.06%) had Gram-positive morphology (Table 2).

### 3.3. Bacterial Confirmation with VITEK 2 Compact

After obtaining the 24 h pure cultures of each bacterial isolate, the Vitek 2 Compact system was used for the final identification.

The most isolated bacterial species from rectal swabs were *Escherichia coli* (43/106, 40.57%), *Salmonella* spp. (18/106, 16.98%), *Staphylococcus vitulinus* (13/106, 12.26%), and *Staphylococcus aureus* (7/106, 6.61%). On the other hand, the species identified from the nasal samples were mostly represented by *Escherichia coli* (42/115, 36.52%), *Staphylococcus lentus* (27/115, 23.48%), and *Staphylococcus aureus* (9/115, 7.82%) (Table 3).

## 4. Discussion

This study reports the data regarding harvesting quota dynamics and detection of potentially pathogenic bacteria between 2017 and 2021 from the European fallow deer (*Dama dama*) residing in three hunting grounds in Western Romania. The harvesting data for bucks, does, and yearlings were individually recorded yearly at every location, thereafter centralizing the three categories of deer in two groups as follows: Group M (bucks) and Group F (does, yearlings). The does and yearlings were classified as one group due to their solidary social behavior, with the latter remaining by their mother’s side until sexual maturity, after which they are considered adults. 

During the five-year period of study, 1881 specimens of fallow deer were hunted. Harvesting quotas in the Socodor location (A) indicated yearly variations, ranging between 238 and 345 fallow deer. In the second location, Chișineu Criș–Sălișteanca (B), harvesting quotas registered a minimum and maximum of 52 and 63, respectively. Lastly, in the Nadăș hunting ground (C), the recorded harvesting quotas varied between 20 and 40 animals. In hunting ground A, an obvious discrepancy between the harvested specimens of the two study groups was noticed, especially in 2017–2019. Because the legislation does not specify the proportions or hunting rates for the deer categories, the possibility of an existing unwritten rule was investigated. This hypothetical measure would mitigate intensive hunting of bucks or, conversely, the more vulnerable does with yearlings in order to avoid an unwanted imbalance in the sexes. Thus, in hunting ground A (Socodor), a negative correlation was found, with no statistically significance between the hunting bags of group M (bucks) and group F (females and yearlings) (r(8) = −0.33; *p* > 0.05). This group is the largest in Europe and highly frequented for its trophy bucks. As shown in Figure 2, the number of hunted bucks has a yearly increase during the investigated period, while the hunting activity corresponding to group F has an increasing trend until 2019, after which a descending trend is noticed. This is believed to ensure the continued thriving of the local fallow buck population through the reproductive and nurturing roles of the females. Therefore, this negative correlation, even though it is weak and statistically insignificant, supports the prestigious reputation of this hunting fund, renowned for its size and the trophy bucks it offers. On the other hand, in the smaller, less internationally recognized hunting ground B (Chișineu Criș–Sălișteanca), the hunting pattern focused on maintaining the sex balance within the deer population. This is evidenced by the strong positive correlation between the hunting dynamics of the two groups, r(8) = 0.95; *p* < 0.05. 

Other national studies that have monitored local fallow deer populations have attributed the discrepancy in sexes to poaching. However, this was not the aim of this research; moreover, there are no accessible data regarding the surviving herd’s disproportion in sexes. In this regard, a national study conducted a retrospective analysis over the dynamics of both the evolution and hunting of the European fallow deer populations in Romania. The researchers observed a disproportion in the harvest quotas between the sexes, despite the fact that the calves, both males and females, are born in approximately equal proportions. Moreover, it is considered that bucks cannot be more exposed than females to natural losses, but their findings suggested the contrary. Unfortunately, the researchers have suggested that this disproportion in sexes is the result of poaching of bucks for trophies, which is an ongoing problem in Romania, due to high levels of corruption. Additionally, they suspected that the real deer population were being deliberately exaggerated in the interest of obtaining increasingly high harvesting quotas in the following years [23]. In another study by Bijl et al. (2022), a large decrease in fallow deer, roe deer, and red deer populations was observed in Romania, which is believed to have been caused by high levels of poaching, unreliable data, or political and economic changes, leading to poor game management [1,28]. Another national study monitored a population of *Dama dama* in Buzău county, Romania, since their introduction in the area. The author mentioned that 39 carcasses were collected during the period of 1972–2003, and between one and seven animals were being shot annually during organized hunting parties. Moreover, variations and discrepancies of the harvesting quotas were noticed between the sexes (i.e., two males and five females in 1975; four males and three females in 1981) as well. It was also mentioned that the residing population of fallow deer was mostly affected by poaching, or by migration as there were consecutive years in which hunting activities were halted, yet the herd was decreasing numerically, and the ratio of the sexes within the population were always in favor of the females [29]. In the present work, the carcasses resulted from legal hunting activities. Thus, the collected data suggest that the hunting dynamics are driven by the desire to prevent population imbalances and to maintain the prestige of a hunting group recognized at the European level, as it was in the case of the Socodor hunting ground.

In Europe, studies have proven that the fallow deer is able to increase their population and reach higher densities despite the greater hunting pressure, when compared to the red and roe deer [30]. Moreover, the fallow deer can easily thrive on lower-quality resources thus successfully adapting to areas impacted by anthropogenic activities [31,32]. This could also be an explanation for the ascending trend of the harvesting amount, observed over the period of our study, including the years that followed after a lower rate of harvest was registered. This hypothesis is in accordance with the study of Bijl et al. (2022), who showed that over the past decades, on the European continent, the fallow deer hunting bag increased 5-fold, compared to the roe deer and red deer, which registered an increase of 1.6-fold and 1.7-fold, respectively [1].

The occurrence of potentially pathogenic bacteria in wild ruminants in Europe is extremely variable. This study revealed that most of the bacteria isolated from European fallow deer can be potential health hazards to both humans and animals (e.g., *Escherichia*, *Staphylococcus*, *Salmonella*, *Listeria*, *Enterococcus*, *Yersinia*). Moreover, the prevalence recorded in the research is in accordance with the values that were reported by other authors, which confirms that wild cervids can act as reservoirs, amplifiers, or be a link in zoonotic diseases [15,21]. In fact, while wildlife can pose a risk to humans and domestic animals, the majority of interactions between microbes and wildlife are innocuous and necessitate little concern [33].

A high prevalence in the microorganisms belonging to the family *Enterobacteriaceae* (66.59%) was reported by Gnat et al. (2015), while studying stool samples from wild cervids from three countries. Their research analyzed 120 fecal samples, concluding that the dominant potentially pathogenic isolated species was *Escherichia coli* (100%; 120/120) [15].

In Poland, several studies have since expanded the knowledge about the frequency values of *E. coli* in the fallow deer. During the period of 2017 to 2018, in Northeastern Poland, several wild cervids were sampled through rectal swabs. The results showed that *E. coli* had a prevalence of 42.85% (9/21) in fallow deer, 39.55% (53/134) in roe deer, and 56.70% (55/97) in red deer [19]. Another study in Poland by Wasyl et al. (2017) screened wild ruminant fecal samples for *E. coli* and successfully isolated the pathogen in fallow deer (21/24), red deer (176/225), and roe deer (64/76) at frequencies of 87.5%, 78.22%, and 84.21%, respectively [34].

Similar results were reported in a previous study conducted by Velhner et al. (2018). The research team analyzed fecal samples (*n* = 106) from cervids living in a community with other wild animals, from four hunting grounds located in the Republic of Serbia. A total of 101 isolates of *E. coli* were obtained at a prevalence of 95.28% [35]. 

Moreover, in Portugal, Dias et al. (2019) identified pathogenic strains of *Escherichia coli* in wild populations of fallow deer inhabiting areas impacted by anthropogenic activities. Therefore, out of 50 isolates of *E. coli*, 68% (34/50) were pathogenic strains, mostly Shiga toxin-producing *E. coli* (STEC) [17]. STEC strains can be transmitted by direct or indirect contact, although they are mostly transmitted through the consumption of food products and water contaminated with feces [36]. Ruminants, including deer, are asymptomatic STEC hosts because of their lack of Shiga toxin vascular receptors [37,38].

Other isolated microorganisms of the *Enterobacteriaceae* family that could pose a threat belong to the genus *Salmonella*. Compared to the obtained results, a study by Pista et al. (2022) in Portugal showed a lower prevalence of *Salmonella* spp., with only in 1 out of 12 (8.3%) red deer stool samples testing positive, while the pathogen was absent in 50 stool samples of *Dama dama* [18]. On a different continent, Gorski et al. (2011) confirmed the presence of *Salmonella enterica* in 104 fecal samples collected from wild cervids in California [39]. 

Contrary to the findings of this research, Gnat et al. (2015) examined 120 wild deer fecal samples and all of them tested negative for *Salmonella*. Moreover, according to other researchers, this microorganism is not detected in wild ruminant feces or is rarely, if ever, detected [40,41]. In general, *Salmonella* is are the main cause of diarrhea and are extensively disseminated in the living habitat of wildlife [42].

Game meat is considered to be a potential source of human listeriosis. In addition to this, *Listeria monocytogenes* has been frequently isolated from areas impacted by anthropogenic activities, especially near farm areas, and is a probable main cause of infection and mortality within the *Cervidae* family. Because of the high survivability rate in the environment, *Listeria* pathogens can be transmitted between farms and wild habitats by asymptomatic carrier animals thus contaminating the wildlife [15,43]. Similarly, in their study, Obwegeser et al. (2012) obtained 11 isolates of *L. monocytogenes* from wild ruminants, showing a prevalence of 4.6% [8]. In other studies, conducted in Germany and Italy, the pathogen in question was recovered from both the meat and carcasses of game ungulates, roe deer, red deer, and wild boars [44,45]. On the other hand, Weindl et al. (2016) collected 45 wild deer samples in three different regions of Germany and Austria in order to identify *L. monocytogenes* and registered a higher prevalence of 17.77%, with 8 out of 45 deer being positive to this microorganism [46].

Although *Staphylococcus aureus* is known as a commensal species, it is also a popular pathogen with significant impact on livestock industry and public health [8]. A study conducted by Mateus-Vargas et al. (2022) in Germany determined the prevalence of *Staphylococcaceae* bacteria from the nasal samples of European fallow deer, red deer, and roe deer, and their results showed that the prevalence of the isolated staphylococci for the examined wild cervids were 21.8% (19/87), 22.7% (5/22), and 11.5% (14/122), respectively. The most frequently isolated species was *S. aureus*, followed by *S. saprophyticus* and *S. epidermidis* [47]. Another study conducted in Poland reported a low prevalence of 3.33% (4/120) in wild deer originating fecal *Staphylococcus aureus* [15]. In their research, Monecke et al. (2016) collected samples from various wildlife species located in Germany, Austria, and Sweden, including European fallow deer, roe deer, red deer, and reindeer. From fecal matter, nasal and rectal swabs, the authors managed to isolate 155 strains of *S. aureus* [48]. This species may induce a less severe type of illness in deer than in other domestic animals, as none of the studies have reported casualties caused by it. 

Regarding other identified staphylococcal species, this research could be compared to only one research on wild cervids that determined the occurrence of *Staphylococcus* spp. in roe deer, isolating one of each strain, *S. xylosus* and *S. vitulinus* [49].

Wild deer have been confirmed to be hosts for the *Enterococcus* species in other studies. As a species of epidemiologic importance, *Enterococcus faecium* was the only one identified in this study. These results are in accordance to Hamarova et al. (2021), who isolated *E. faecium* from the wildlife of Slovakia, at a prevalence of 6.4% [50]. Similar findings were stated by other authors, who successfully isolated 12 strains of this bacterial species from feces of red deer located in Hungary, Slovakia, and Poland [15]. Another study in Sweden conducted by Lillehaug et al. (2005) analyzed 50 wild cervid stool samples and recovered *E. faecium* at a higher prevalence of 14% [51]. The *Enterococcus* species, commonly considered with low pathogenicity, have become significant sources of nosocomial and community infections due to their ability to acquire virulence traits. Additionally, they are inherently resistant to many antibiotics and can transfer this resistance to other microorganisms [51,52]. Given the scarcity of published research on *Enterococcus faecium* in deer, it is difficult to assess their spread in *Cervidae* family, including the fallow deer.

Despite the fact that *Pseudomonas oleovorans* is usually not associated with diseases in humans and animals and is considered an environmental contaminant from industrial wastewaters, a study has shown that it could cause septic infections in children [53]. Since no other studies related to the identification of *Pseudomonas oleovorans* in fallow deer have been found, this research can be considered the first study in Romania and Europe that has identified *Pseudomonas oleovorans* in wild *Dama dama*.

Moreover, other species isolated from the fallow deer in this research, such as *Enterobacter* spp., *Kocuria kristinae*, *Rothia dentocariosa*, *Providencia rettgeri*, *Aerococcus viridans*, *Leuconostoc mesenteroides* ssp. *cremoris*, and *Gemella* spp., are known as common inhabitants of animals and human skin and mucosae. Several authors have affirmed that these species present no epidemiological importance in wildlife, as there is only evidence that they cause nosocomial infections in immunocompromised and chronically affected humans [15,53,54,55,56].

## 5. Conclusions

As a species cherished for its ecological significance and economic value in hunting, *Dama dama* populations in Romania, as well as other European nations, require ongoing surveillance. Moreover, *Dama dama* may harbor zoonotic pathogens, emphasizing the importance of a One Health approach and the need for regular monitoring of this ruminant. As fallow deer populations, particularly in Europe, continue to expand, identifying the zoonotic diseases from them becomes increasingly essential for a better understanding of how wildlife has an important role in environmental contamination and the transmission of pathogens to human consumers through the game meat. 

## Figures and Tables

**Figure 1 microorganisms-12-01236-f001:**
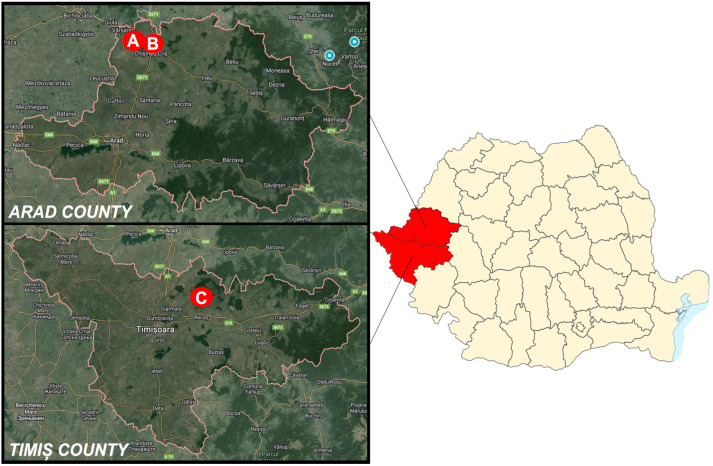
The sites of the hunting grounds, Socodor (A), Chișineu Criș–Sălișteanca (B) (Arad county), and Nadăș (C) (Timiș county). Map source: Google Maps.

**Figure 2 microorganisms-12-01236-f002:**
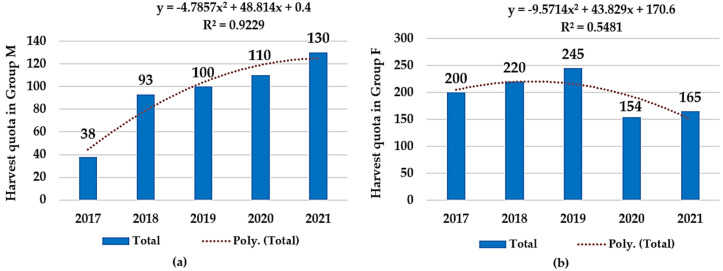
A comparison in harvesting quota dynamics between 2017 and 2021 in Hunting Ground Socodor: (**a**) Group M—ascending trend; (**b**) Group F—slight descending trend.

**Figure 3 microorganisms-12-01236-f003:**
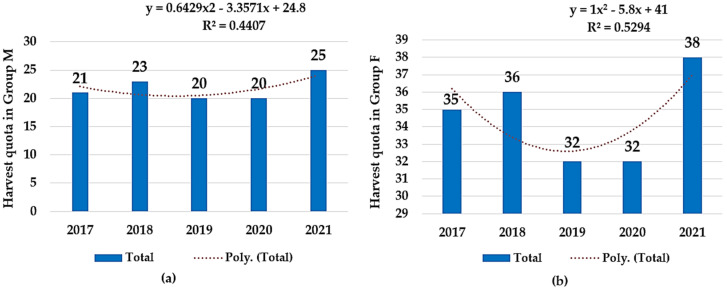
A comparison in harvesting quota dynamics between 2017 and 2021 in Hunting Ground Chișineu Criș–Sălișteanca: (**a**) Group M—slightly ascending trend; (**b**) Group F—slightly ascending trend.

**Figure 4 microorganisms-12-01236-f004:**
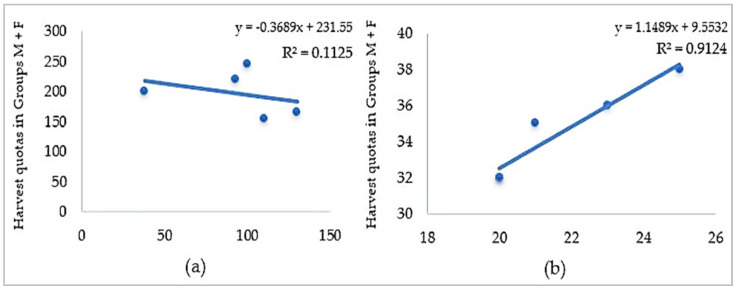
Relationship between the harvesting quota dynamics of Group M and Group F between 2017 and 2021: (**a**) in Hunting Ground A; (**b**) in Hunting Ground B.

**Table 1 microorganisms-12-01236-t001:** The total annual harvesting bag between 2017 and 2021 across all three hunting reserves, Socodor (A), Sălișteanca (B), and Nadăș (C).

Category	Harvesting Quotas				
	2017 No. (%)	2018 No. (%)	2019 No. (%)	2020 No. (%)	2021 No. (%)
Group M	38	93	100	110	130
Group F	200	220	245	154	165
TOTAL (A)	238 (75.80)	313 (79.44)	345 (79.99)	264 (75.86)	295 (74.12)
Group M	21	23	20	20	25
Group F	35	36	32	32	38
TOTAL (B)	56 (17.83)	59 (14.97)	52 (12.17)	52 (14.94)	63 (15.83)
Group M	5	7	8	9	10
Group F	15	15	22	25	30
TOTAL (C)	20 (6.37)	22 (5.59)	30 (7.02)	32 (9.20)	40 (10.05)
TOTAL (A + B + C)	314 (100)	394 (100)	427 (100)	348 (100)	398 (100)

No., Number; Group M, Harvested Bucks; Group F, Harvested Does and Yearlings; A, Socodor Hunting Ground; B, Chișineu Criș–Sălișteanca Hunting Ground; C—Nadăș Hunting Ground; A + B + C, Total Harvesting Quotas from the three hunting funds.

**Table 2 microorganisms-12-01236-t002:** The complete count of bacterial strains on selective culture media.

Sample Origin	No. Positive Samples/Examined (%)				Gram Staining (*n* = 221)	
	L	Ra	Ch	Ox	No. of GNB (%)	No. of GPB (%)
Rectal	46/120 (38.33)	19/120 (15.83)	38/120 (31.66)	3/120 (2.50)	41 (35.04)	64 (61.54)
Nasal	47/120 (39.16)	5/120 (4.16)	58/120 (48.33)	5/120 (4.16)	76 (64.96)	40 (38.46)
Total	93/240 (38.75)	24/240 (10)	96/240 (40)	8/240 (3.33)	117 (100)	104 (100)

L, Levine; Ra, Rambach; Ch, Chapman; Ox, Oxford; No., Number; GNB, Gram-Negative Bacteria; GPB, Gram-Positive Bacteria.

**Table 3 microorganisms-12-01236-t003:** Identification and prevalence of bacterial isolates.

Sample Origin	Identified Strains	No. of Isolated Bacteria (*n* = 221)	Prevalence (%)
Rectal (*n* = 120)	*Escherichia coli*	43	40.57
	*Salmonella* spp.	18	16.98
	*Staphylococcus vitulinus*	13	12.26
	*Staphylococcus aureus*	7	6.61
	*Staphylococcus lentus*	5	4.72
	*Enterobacter* spp.	3	2.83
	*Enterococcus faecium*	3	2.83
	*Staphylococcus xylosus*	3	2.83
	*Rothia dentocariosa*	3	2.83
	*Aerococcus viridans*	3	2.83
	*Kocuria kristinae*	2	1.89
	*Staphylococcus hominis*	1	0.94
	*Pseudomonas oleovorans*	1	0.94
	*Leuconostoc mesenteroides ssp. cremoris*	1	0.94
	Total	106	100
Nasal (*n* = 120)	*Escherichia coli*	42	36.52
	*Staphylococcus lentus*	27	23.48
	*Staphylococcus aureus*	9	7.82
	*Aerococcus viridans*	5	4.35
	*Listeria monocytogenes*	5	4.35
	*Salmonella* spp.	4	3.48
	*Enterobacter* spp.	4	3.48
	*Kocuria kristinae*	4	3.48
	*Staphylococcus vitulinus*	4	3.48
	*Gemella* spp.	3	2.61
	*Staphylococcus xylosus*	3	2.61
	*Staphylococcus sciuri*	2	1.73
	*Pseudomonas oleovorans*	1	0.87
	*Providencia rettgeri*	1	0.87
	*Staphylococcus hominis*	1	0.87
	Total	115	100

No., Number; Spp., Species.

## Data Availability

The original contributions presented in the study are included in the article, further inquiries can be directed to the corresponding author.

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
