# Peer review of "Hunting Dynamics and Identification of Potentially Pathogenic Bacteria in European Fallow Deer (Dama dama) across Three Hunting Reserves in Western Romania"

_microorganisms, 2024, doi:10.3390/microorganisms12061236_

Round 1
Reviewer 1 Report
Comments and Suggestions for Authors
1. In line 48, add a space before "Simultaneously".
2. In line 107, add a ")" after "bacteria".
3. In Figure 4, what is the significance of the harvesting quotas correlation between group M and group F? Why calculate the correlation?
4. Lines 197-200 repeat lines 201-205.
5. In lines 230-235, is the disproportion in sexes of harvesting quotas because group F is more vulnerable to hunting? Is there any data showing the disproportion in sexes in the surviving herd?
6. In lines 242-249, can the references from 1972 to 2003 indicate the situation in recent years?
Comments on the Quality of English LanguageModerate editing of English language required.
Reviewer 2 Report
Comments and Suggestions for Authors
Dear authors
This is a well-written scientific study. English language and grammar are adequate and sound understandable.
The topic is quite interesting as it tries to link bacterial presence of zoonotic importance with harvesting and possible overpopulation.
L97-98: To collect and transport the samples which swabs and transport medium did you use? Please add
L98-99: How much time interfered between sampling, cooling and transport to vet faculty? Please define
L102: Please use/add an adequate citation to quote the lab methods as well as all materials used
L105: Did you use any special culture media e.g for Salmonella, Mycobacteria ?
L106: Please apart from the companies elaborate on the media quality (e.g blood agar, McConkey)
L114-115: Although Vitek 2 is very reliable it would have added value if you had further sequence the most important zoonotic bacteria by MALDI-TOPF or similar equipment. Would it be a possibility to expand your analysis?
In your introduction and discussion, the part of the microorganisms could be improved if you use some more updated publications. I quote some of them below:
Odyniec, M., & Bancerz-Kisiel, A. (2022). Assessment of the Role of Free-Living and Farmed Fallow Deer (Dama dama) as A Potential Source of Human Infection with Multiple-Drug-Resistant Strains of Yersinia enterocolitica and Yersinia pseudotuberculosis. Pathogens, 11(11), 1266.
Tîrziu, E., Bulucea, A. V., Imre, K., Nichita, I., Muselin, F., Dumitrescu, E., ... & Cristina, R. T. (2023). The Behavior of Some Bacterial Strains Isolated from Fallow Deer Compared to Antimicrobial Substances in Western Romania. Antibiotics, 12(4), 743.
Gheibipour, M., Ghiasi, S. E., Bashtani, M., Montazer Torbati, M. B., & Motamedi, H. (2023). Tannase-producing bacteria isolated from the rumen of Fallow deer (Dama dama): Livestock potential feed additives. Biological Journal of Microorganism, 12(48), 27-40.
Szczerba-Turek, A., Siemionek, J., Socha, P., Bancerz-Kisiel, A., Platt-Samoraj, A., Lipczynska-Ilczuk, K., & Szweda, W. (2020). Shiga toxin-producing Escherichia coli isolates from red deer (Cervus elaphus), roe deer (Capreolus capreolus) and fallow deer (Dama dama) in Poland. Food microbiology, 86, 103352.
Di Blasio, A., Varello, K., Vitale, N., Irico, L., Bozzetta, E., Goria, M., ... & Dondo, A. (2019). Animal tuberculosis in a free-ranging fallow deer in northwest Italy: a case of “lucky strain survival” or multi-host epidemiological system complexity?. European journal of wildlife research, 65, 1-9.
Reviewer 3 Report
Comments and Suggestions for Authors
Dama Dama may carry zoonotic pathogens, and regular monitoring of zoonotic diseases as the number of fallow deer increases, especially in Europe, is becoming increasingly important for better understanding how wildlife plays an important role in environmental pollution and transmission. Pathogens are transmitted to human consumers. I think this article has some help and guidance for better prevention and control of zoonotic infectious diseases.
1) Between 2017 and 2021, What is the basis for selecting this time period?
2) he collected swabs with samples (n=240) underwent classic bacteriological techniques in order to isolate different species of bacteria. Why haven't other bacteria been isolated as indicators?
3) In figure 2, Why is there no error line? Error lines are commonly used in statistical or scientific data to indicate potential errors or the degree of uncertainty relative to each data marker in the series.
Comments on the Quality of English LanguageIt is best to have a moderate level of language improvement.
Round 2
Reviewer 1 Report
Comments and Suggestions for Authors
The analysis of the correlation between hunting male and female deer, and then extrapolates from the sex ratio imbalance is the reason for hunting, it should be explained clearly in the discussion.
Author Response
Dear Reviewer,
Thank you for your insightful comments. We have carefully considered your suggestion to clarify the discussion section and have made the necessary modifications. The revised manuscript now includes a clear explanation of the correlation between hunting male and female deer, and how it relates to the sex ratio imbalance (Lines 232-251). We hope this addresses your concern and improves the overall clarity of the paper.
Thank you again for your time and expertise.